# The Impact of Test Anxiety and Cognitive Stress on Error-Related Brain Activity

**DOI:** 10.3390/bs16010025

**Published:** 2025-12-22

**Authors:** Zhenni Jin, Fangfang Long, Hua Wei

**Affiliations:** 1Department of Psychology, Suzhou University of Science and Technology, Suzhou 215000, China; jinzhen0421@gmail.com; 2School of Psychology, Guizhou Normal University, Guiyang 550025, China; alongfangfang@163.com

**Keywords:** test anxiety, cognitive stress, ERN, Flanker task

## Abstract

Test anxiety is considered to affect individuals’ cognitive control and task performance, particularly in error monitoring. While previous research has explored the relationship between anxiety and cognitive performance, this study aims to investigate the impact of test anxiety and cognitive stress on error processing, focusing on changes in error-related negativity (ERN). Participants were divided into high test anxiety (HTA) and low test anxiety (LTA) groups based on their scores on the Test Anxiety Scale (TAS). Cognitive stress was induced by administering the Raven’s Standard Progressive Matrices test, accompanied by instructions that emphasized score comparison with others. Participants completed the Subjective Test Anxiety Scale (STAS), the Short State Anxiety Inventory (SSAI), and a Flanker task before and after the stress manipulation. The effectiveness of the stress manipulation was confirmed by significant increases in STAS and SSAI scores and changes in behavioral performance. EEG data were recorded to analyze ERN, correct-response negativity (CRN), and ΔERN (ERN minus the CRN) amplitudes. HTA individuals exhibited a trend toward larger ERN amplitudes than LTA counterparts, indicating heightened sensitivity to errors. However, no significant changes in ERN amplitudes were observed between pre- and post-stress conditions. CRN and ΔERN amplitudes also showed no significant differences across anxiety groups or stress conditions. ERN changes appear to be more closely related to trait test anxiety than to transient stress.

## 1. Introduction

Anxiety is an unpleasant emotional or mental state experienced by individuals, encompassing various types, including test anxiety. Among students, test anxiety is a widespread and significant issue, with 22% reporting symptoms ([38]). It involves the worry and emotional and behavioral responses triggered by concerns about potential poor performance in test settings ([86]). Notably, test anxiety has also been associated with heightened motivation in cognitive stress contexts, which may modulate its cognitive effects ([2]). According to Attentional Control Theory, test anxiety impairs an individual’s attentional control function and leads to cognitive decline ([19]; [24]; [25]).

To better understand these cognitive impairments, research has increasingly turned to neural markers of performance monitoring, most prominently the error-related negativity (ERN). ERN is a key neural indicator in monitoring the process of error detection, evaluation, feedback, and behavioral adjustment, effectively measuring the individual’s error-processing ([6]; [37]; [74]). It usually occurs 0–150 ms after an error response, and the faster the required response rate, the greater the likelihood that an error will be made, leading to an increase in the ERN amplitude ([26]; [30]). The ERN reflects the role of the anterior cingulate cortex in monitoring and processing errors ([5]; [18]; [27]; [85]). Individuals with anxiety disorders tend to be more sensitive to error-related information, leading to an increase in ERN amplitude when they become aware of mistakes, as this response captures their heightened focus on these errors ([15]; [34]; [45]). Anxiety predisposes individuals to excessively focus on potential negative outcomes, thereby amplifying their neural response to mistakes.

Building on this, the ERN has been proposed as a biomarker for anxiety risk. Previous studies have shown that anxiety-induced cognitive decline is manifested as elevated ERN amplitudes ([45]; [49]). It has been observed in individuals with high trait anxiety, generalized anxiety disorder, obsessive–compulsive disorder, and social anxiety disorder ([11]; [17]; [33]; [59]). However, not all findings are consistent. Recent studies have raised questions about the reliability of ERN as a marker for anxiety. For example, [14] ([14]) reported no significant association between ERN and trait anxiety across multiple analytical methods in a non-clinical sample. [68] ([68]) also suggested that the reported link between ERN and anxiety may be inflated due to publication bias. These findings suggest that the relationship between ERN and anxiety may be more complex than initially assumed, potentially moderated by factors such as task design, anxiety subtypes, or individual differences.

Despite extensive prior research indicating cognitive deficits associated with test anxiety ([25]; [67]), there remains a gap in studies specifically examining whether test anxiety affects the ERN. When studying whether test anxiety affects an individual’s ERN, a key issue to consider is whether the impairment of cognitive function due to test anxiety arises from situational anxiety or from the trait itself, a point that remains debated.

On one hand, extensive research has demonstrated that, even in the absence of a cognitive stress situation, test anxiety undermines cognitive functioning. This impact manifests in several aspects, including reduced working memory capacity ([72]), impaired attentional control ([79]; [80]), and the direct interference of emotional components with cognitive processes ([12]; [80]). Studies have indicated that test anxiety depletes valuable cognitive resources, making it difficult for individuals to filter out distracting information, leading to decreased processing efficiency rather than changes in task accuracy ([55]; [72]). This impairment has been confirmed through behavioral experiments and neurophysiological research, showing lower performance efficiency and abnormal brain activity in individuals with test anxiety during attention and inhibitory control tasks ([3]). Ultimately, these cognitive impairments can affect academic performance and reduce overall learning outcomes.

In contrast, some researchers have found that test anxiety impairs cognitive function for individuals only under a cognitive stress situation, especially in tasks requiring attentional control ([61]; [77]). Research has shown that high test anxiety (HTA) individuals show reduced attentional control only under the threat of performance evaluation, resulting in poorer task performance ([60]; [77]). For example, [9] ([9]) used working memory tasks and found that HTA individuals exhibited reduced working memory capacity under stress conditions. However, in the absence of stress, their performance did not significantly differ from that of low test anxious (LTA) individuals. According to this view, under stressful situations, HTA individuals are more likely to become distracted and focus on threatening stimuli unrelated to the task, rather than on the task itself. This leads to longer reaction times (RTs) and a decrease in their rate of correctness ([60]; [78]). These studies emphasize the role of cognitive stress situations in the cognitive deficits of individuals with test anxiety.

This controversy over the effects of context and anxiety itself on ERN is not only present in test anxiety. It also permeates the broader literature on anxiety and the ERN. On one hand, several studies indicating that anxiety itself affects ERN ([22]; [44]; [58]; [64]). To exemplify this, obsessive–compulsive disorder has often been cited as a representative clinical example within the anxiety spectrum. We reference obsessive–compulsive disorder here because, much like test anxiety, it is characterized by excessive worry and hyperactive performance monitoring. Consistent with this, it has been shown that patients with obsessive–compulsive disorder exhibit an overactive error monitoring system even in the absence of a cognitive stress situation, especially showing a significant increase in ERN ([22]). Furthermore, [44] ([44]) provided additional evidence indicating that behavioral inhibition in childhood is associated with heightened response monitoring, as measured by ERN, during adolescence. This elevated ERN, observed even in the absence of a cognitive stress situation, is linked to an increased risk of clinically significant anxiety disorders later in life. However, there is also evidence that anxiety requires a cognitive stress situation to have an effect on ERN. For instance, [64] ([64]) found that ERN enhancement was more pronounced in individuals with high trait anxiety under stressful situations, suggesting that high trait anxiety amplifies the effects of error monitoring under stress. Similarly, [7] ([7]) used an attentional network testing paradigm and found that early childhood anxiety tendencies significantly enhanced ERN amplitudes in preschoolers, but only under cognitive stress.

Given these complexities, clarifying the nature of test anxiety is critical to explaining changes in ERN, as giving a cognitive stress situation may affect cognitive function. Prior studies on other forms of anxiety and ERN suggest a similar phenomenon, in which a cognitive stress situation may be necessary for impairments to appear ([58]; [64]). Despite extensive ERN research in clinical and trait anxiety, to the best of our knowledge, no study has empirically tested this specific interaction within the domain of test anxiety. Consequently, it remains unexplored whether these neural mechanisms are driven by the trait itself or by situational stress. To address these controversies and further our understanding of the effects of test anxiety, we delve into the role of ERNs in test anxiety. This approach not only improves our understanding of the mechanisms that influence test anxiety but also provides new perspectives for developing effective interventions.

This study aimed to examine the combined effect of test anxiety and cognitive stress situation on error processing. In this experiment, we created a test-like cognitive stress situation in the laboratory, requiring participants to complete the Flanker task both before and after the induction of stress ([76]). The Flanker task serves as an established tool to measure attentional control and error processing ([20]; [29]). Indeed, a systematic review of ERN and correct-response negativity (CRN) in anxiety disorders included 66 papers, 51 of which used the Flanker task to measure the ERN ([49]). Because individuals with HTA are more sensitive to error detection in stressful situations, we hypothesized that HTA individuals would show enhanced ERN amplitudes relative to LTA individuals, and that this ERN increase would be significantly larger under the cognitive-stress condition.

## 2. Methods

### 2.1. Participants

Undergraduate students from Suzhou University of Science and Technology were recruited for this study through an online survey and offline posters. The online survey was administered via the WenJuanXing platform, https://www.wjx.cn/ (accessed on 1 March 2024). The Test Anxiety Scale (TAS) was used to assess test anxiety levels ([66]). According to [54] ([54]), TAS scores of 12 or below indicate low levels of test anxiety, scores between 13 and 19 reflect moderate anxiety, and scores of 20 or above represent high levels of test anxiety. Based on this, only individuals with high or low scores were included in the HTA and LTA groups. This classification method has been widely adopted and validated in previous research ([21]; [54]; [77]; [83]; [86]). A total of 388 questionnaires were collected, of which 252 met the scoring criteria. Due to scheduling conflicts or unwillingness to participate in the electroencephalography (EEG) experiment, only 94 people agreed to participate in the experiment. To ensure a sufficient number of error trials for the ERN analysis, data from five participants who committed fewer than six errors were excluded ([57]). Ultimately, the final sample included 44 LTA participants (23 females; *Mean* [*M*] = 19.82, *Standard Deviation* [*SD*] = 1.41) and 45 HTA participants (25 females; *M* = 19.53, *SD* = 1.27). Participants were aged 18–23 years, and data collection took place between 1 March and 11 May 2024. A power analysis was conducted using G*Power 3.1.9.7 ([28]) to determine the required sample size for a 2 (group: HTA and LTA; between-subject) × 2 (condition: pre-stress and post-stress; within-subject) mixed-design ANOVA. Assuming a medium effect size (f = 0.25), α = 0.05, power = 0.95, and a correlation among repeated measures of 0.5, the analysis indicated that a total of 54 participants would be sufficient. Thus, the final sample size (*n* = 89) exceeded the minimum requirement for adequate statistical power. The experimental protocol was approved by the Ethics Committee of the Academic Committee of Suzhou University of Science and Technology. All participants provided informed consent and received compensation of 50 CNY (about USD 6.95) upon completing both parts of the study: the online questionnaires and the EEG recording session.

### 2.2. Materials

#### 2.2.1. TAS

The TAS is an extensively utilized instrument for assessing test anxiety levels among students ([66]). The scale consists of 37 items, of which six (items 3, 15, 26, 27, 29, and 33) are reverse-scored. The scale has demonstrated high internal consistency in previous studies (Cronbach’s α = 0.87, *n* = 1996), supporting its reliability in the Chinese student population ([8]; [71]). Items reflect worry, self-doubt, and test-related intrusive thoughts. For example, item 1 reads: “Before an important test, I often think others are much smarter than I am,” to which participants respond either agree or disagree.

#### 2.2.2. Subjective Test Anxiety Scale (STAS)

Participants’ momentary test anxiety was assessed using the STAS, a single-item self-report measure. The item asked, “Please select your current level of test anxiety,” with responses rated on a 5-point Likert scale: 1 (“completely relaxed”), 2 (“relatively relaxed”), 3 (“neutral”), 4 (“relatively anxious”), and 5 (“very anxious”). This scale was designed to capture participants’ real-time subjective perceptions of test anxiety both before and after the stressor. Higher scores reflect greater state anxiety.

#### 2.2.3. Short State Anxiety Inventory (SSAI)

The SSAI was employed to assess participants’ current state of anxiety and evaluate changes in anxiety levels. The short version consists of six items, focusing on both positive (e.g., “I feel calm,” “I feel content”) and negative emotional states (e.g., “I feel tense,” “I feel worried”). Responses are rated on a 4-point Likert scale ranging from 1 (“not at all”) to 4 (“very much”). Participants were instructed to respond based on their immediate feelings. Items 1, 4, and 5 were reverse-coded. The SSAI demonstrated good internal consistency (Cronbach’s α = 0.81, *n* = 671), supporting its suitability for this research ([84]).

#### 2.2.4. Raven’s Standard Progressive Matrices (RSPM)

The RSPM, developed by J.C. Raven ([62]), is a non-verbal intelligence test that evaluates abstract reasoning through pattern completion tasks. In this study, we used 24 items from sections D and E of the RSPM, which are characterized by higher difficulty levels. The task was used to induce a cognitive stress situation related to tests. To simulate a test-like cognitive stress situation, participants were instructed to take the test seriously, with the added information that their scores would be compared to those of other students ([40]).

#### 2.2.5. Flanker Task

The Flanker task used in this study was programmed and presented using E-Prime 2.0 software. As shown in Figure 1, each trial consisted of five white arrows displayed against a black background for 150 ms. Two types of trials were included: congruent and incongruent, each with an equal probability of 50%. In congruent trials, all arrows pointed in the same direction, while in incongruent trials, the central arrow pointed in the opposite direction to the flanking arrows. The trial order was randomized throughout the task. All participants were right-handed and used the “F” and “J” keys to respond, pressing “F” with the left index finger for leftward arrows and “J” with the right index finger for rightward arrows. These keys align with standard typing habits and are thus highly familiar, requiring no special training or adaptation. RTs were recorded as the time elapsed from the onset of the stimuli to the participant’s keypress. If no response was made within 650 ms, the trial was classified as an error. The task then automatically proceeded to the next trial following a blank screen (inter-trial interval) lasting 1500–2000 ms. The Flanker task consisted of 1 practice block (80 trials) followed by 10 experimental blocks, with each block containing 80 trials. At the end of each block, participants received performance feedback based on their accuracy and speed ([23]). If accuracy was below 75%, the feedback displayed was “Your accuracy is too low, please improve”. For accuracy above 90%, participants were informed, “Your response is too slow, please speed up”. If performance fell between these thresholds, the feedback read, “Well done, keep going”.

### 2.3. Procedure

The experiment was divided into three parts: the pre-stress phase, the stress-inducing phase, and the post-stress phase (Figure 1).

**Figure 1 behavsci-16-00025-f001:**
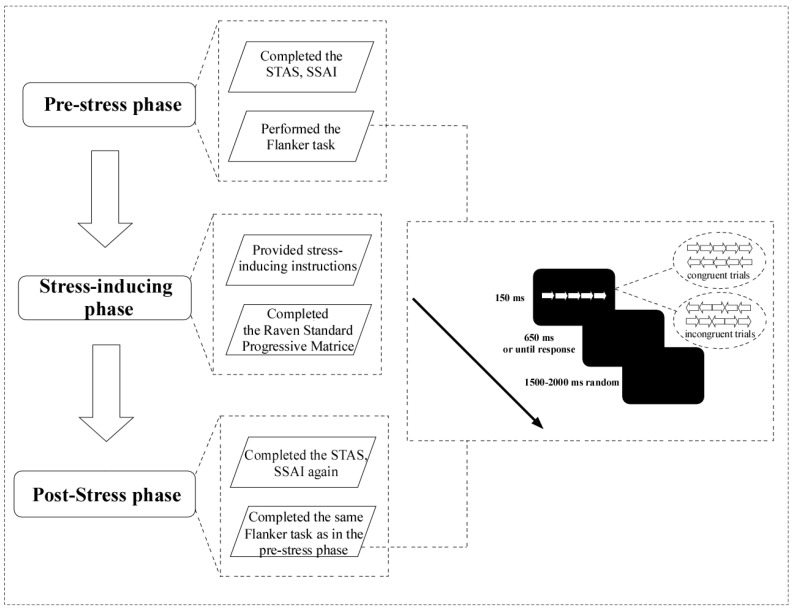
Schematic representation of the experimental procedure (left) and the Flanker task (right). *Notes:* STAS = Subjective Test Anxiety Scale; SSAI = Short State Anxiety Inventory.

In the pre-stress phase, participants first completed the STAS and SSAI. Before completing the Flanker task, participants were asked to adjust their posture approximately 70 cm from a 27-inch computer screen, ensuring that the center of the screen was at eye level. A practice session was conducted to familiarize participants with the task rules. Subsequently, participants completed the Flanker task, which lasted approximately 30–35 min. Rest periods were provided at the end of each block to prevent fatigue.

In the stress-inducing phase, participants were exposed to test-related stress through task instructions and the administration of RSPM. They were explicitly informed that their performance on the tasks would be analyzed in relation to their intelligence levels, highlighting the importance of dedicating maximum effort to the subsequent intelligence test and task sessions. To further heighten the stress, participants were told that their test scores would be compared with those of their peers to assess differences in cognitive abilities. Additionally, a strict 5 min time limit was imposed for completing the RSPM, with answer sheets collected promptly at the end of the allotted time. These procedures were designed to induce a sense of urgency and evaluative threat, simulating a high-stress test phase ([76]).

In the post-stress phase, participants completed the STAS and SSAI again, followed by the same Flanker task as in the pre-stress phase. This task also lasted approximately 30–35 min and followed the same procedure and content as the earlier session.

### 2.4. Electrophysiological Recording and Data Reduction

EEG data were collected using 64 Ag-AgCl scalp electrodes, positioned according to the international 10–20 system (bandpass filter: 0.01–400 Hz; sampling rate: 1000 Hz). The EEG activity was recorded using the Curry 9.2 software and amplified with a SynAmps2 amplifier. The impedance was kept below 10 kΩ before data collection. During recording, the reference electrodes were placed between Cz and CPz, and the ground electrode was positioned at the Afz. Offline, all data was referenced to the average of the left and right mastoids and band-pass filtered between 0.1 and 30 Hz. Independent component analysis was applied to identify and remove artifacts related to ocular and muscle activity. Trials with voltage fluctuations exceeding ±75 μV at any electrode were excluded from further analysis. The EEG data were segmented for each trial beginning 400 ms before the response and continuing for 1000 ms after the response and baseline correction was performed using the interval from −400 to −200 ms ([46]; [58]; [64]). For each subject, the ERN was scored as the mean activity between 0 and 100 ms after error responses at FCz, where error-related brain activity was maximal.

According to Attentional control Theory ([19]; [25]) and prior ERP research, the ERN recorded at frontocentral electrodes, particularly FCz, is considered a reliable neural correlate of prefrontal attentional control. Numerous studies on anxiety and ERN have adopted FCz as the primary site of interest, as this location most consistently captures the maximal ERN amplitude in both clinical and non-clinical populations ([32]; [46]; [57]; [64]). In the present study, topographies similarly confirmed that ERN activity peaked at FCz across both high and low test anxiety groups, consistent with prior findings ([16]; [22]; [58]). Therefore, we selected FCz for ERP analyses in line with our theory-driven hypotheses and existing empirical evidence.

The CRN was scored in the same way on correct trials. The ΔERN was calculated by subtracting the CRN from the ERN, which has been thought to isolate neural activity specific to error processing ([32]). The number of artifact-free trials was sufficient to ensure reliable estimation of ERN and CRN components across all conditions. For the HTA group, the mean number of CRN trials was 519.38 ± 136.83 in the pre-stress condition and 546.42 ± 99.42 in the post-stress condition. The mean number of ERN trials was 79.47 ± 37.37 in the pre-stress condition and 85.40 ± 31.60 in the post-stress condition. For the LTA group, the mean number of CRN trials was 525.02 ± 185.80 in the pre-stress condition and 572.61 ± 143.61 in the post-stress condition. The mean number of ERN trials was 75.48 ± 34.99 in the pre-stress condition and 91.07 ± 35.52 in the post-stress condition. All ERP amplitudes were recorded in microvolts (μV); the unit symbol is omitted in the following results for conciseness.

### 2.5. Statistical Analysis

Participant’s scores on the STAS and SSAI were analyzed separately using a two-way mixed analysis of variance (ANOVAs) with cognitive stress (pre- and post-) as the within-subject factor, and with group (HTA group and LTA group) as the between-subject factor. The RTs and accuracy of the flanker task were analyzed using a three-way mixed ANOVAs, with cognitive stress (pre- and post-) and congruence (congruent and incongruent) as within-subject factors, and group (HTA group and LTA group) as a between-subject factor. The RTs of correct and error in the flanker task were analyzed using a three-way mixed ANOVA, with cognitive stress (pre- and post-) and trial type (correct and error) as within-subject factors, and group (HTA group and LTA group) as a between-subject factor. The ERN and CRN amplitudes were analyzed using a three-way mixed ANOVA, with component (ERN and CRN) and cognitive stress (pre- and post-) as the within-subject factors, and group (HTA group and LTA group) as a between-subject factor. The ΔERN was analyzed using a two-way mixed analysis of variance (ANOVA) with cognitive stress (pre- and post-) as the within-subject factor, and with group (HTA group and LTA group) as the between-subject factor. For all statistical tests, partial eta-squared (*η_p_*^2^) was reported as the effect-size index for ANOVAs, and the significance criterion was set at α = 0.05.

## 3. Results

### 3.1. Self-Reports

#### 3.1.1. STAS

A two-way mixed ANOVA revealed a significant main effect of cognitive stress, *F*(1,87) = 8.48, *p* = 0.005, *η_p_*^2^ = 0.09. STAS scores were higher under cognitive stress (*M* = 2.56, *SD* = 0.93) than under no cognitive stress (*M* = 2.30, *SD* = 0.86). The group also revealed a significant main effect, *F*(1,87) = 31.42, *p* < 0.001, *η_p_*^2^ = 0.27, indicating that the HTA group (*M* = 2.83, *SD* = 0.68) scored higher than the LTA group (*M* = 2.02, *SD* = 0.68). The interaction was not significant, *F*(1,87) = 0.73, *p* = 0.39, *η_p_*^2^ = 0.008.

#### 3.1.2. SSAI

Similarly, a significant main effect of cognitive stress was found, *F*(1,86) = 32.60, *p* < 0.001, *η_p_*^2^ = 0.28. SSAI scores were higher under cognitive stress (*M* = 13.32, *SD* = 3.71) than under no cognitive stress (*M* = 11.55, *SD* = 3.43). The group also revealed a significant main effect, *F*(1,86) = 27.16, *p* < 0.001, *η_p_*^2^ = 0.24, with the HTA group scoring higher (*M* = 14.02, *SD* = 2.90) than the LTA group (*M* = 10.84, *SD* = 2.87). The interaction was not significant, *F*(1,86) = 0.00, *p* = 1.00, *η_p_*^2^ < 0.001.

### 3.2. Behavioral Data

The behavioral data analysis results are presented in Table 1.

#### 3.2.1. RTs

This section analyzed RTs for correct and error trials. A three-way mixed ANOVA on RTs for correct and error trials revealed a significant main effect of cognitive stress, *F*(1,87) = 79.31, *p* < 0.001, *η_p_*^2^ = 0.48, with RTs being shorter under the cognitive stress condition (*M* = 394.62, *SD* = 34.82) than without cognitive stress (*M* = 415.99, *SD* = 36.74). Trial type also revealed a significant main effect, *F*(1,87) = 582.73, *p* < 0.001, *η_p_*^2^ = 0.87, with RTs being shorter for error trials (*M* = 381.20, *SD* = 32.40) compared to correct trials (*M* = 429.41, *SD* = 37.86). A significant interaction between cognitive stress and trial type was observed, *F*(1,87) = 15.77, *p* < 0.001, *η_p_*^2^ = 0.15. Under the condition without cognitive stress, RTs for correct trials (*M* = 442.30, *SD* = 39.62) were significantly longer than the RTs for error trials (*M* = 389.68, *SD* = 37.21). Under the condition with cognitive stress, RTs for correct trials (*M* = 416.52, *SD* = 39.79) were significantly longer than the RTs for error trials (*M* = 372.72, *SD* = 32.42). A paired samples *t*-test revealed that the difference in RTs between correct and error trials was significantly smaller under cognitive stress (*M* = 43.80, *SD* = 20.37) compared to without cognitive stress (*M* = 52.63, *SD* = 22.48), *t*(87) = 3.99, *p* < 0.001, Cohen’s *d* = 0.42. No other main effects or interactions were significant, *F*(1,87) ≤ 0.03, *p* > 0.86.

This section explored RTs for congruence and incongruence trials. A three-way mixed ANOVA on RTs revealed a significant main effect of cognitive stress, *F*(1,87) = 93.02, *p* < 0.001, *η_p_*^2^ = 0.52, with RTs shorter under the cognitive stress (*M* = 410.78, *SD* = 39.82) than without cognitive stress (*M* = 435.27, *SD* = 39.37). Congruence also revealed a significant main effect, *F*(1,87) = 1241.66, *p* < 0.001, *η_p_*^2^ = 0.94, indicating longer RTs in incongruent trials (*M* = 443.81, *SD* = 40.35) than congruent trials (*M* = 402.24, *SD* = 35.82). A significant interaction between cognitive stress and congruence was found, *F*(1,87) = 80.06, *p* < 0.001, *η_p_*^2^ = 0.48. Under the condition without cognitive stress, RTs for congruent trials (*M* = 412.33, *SD* = 37.53) were significantly shorter than the RTs for incongruent trials (*M* = 458.20, *SD* = 42.01). Similarly, under the condition with cognitive stress, RTs for congruent trials (*M* = 392.14, *SD* = 37.67) remained shorter than the RTs for incongruent trials (*M* = 429.43, *SD* = 42.70). A paired-samples *t*-test revealed that the difference in RTs between incongruent and congruent trials was significantly smaller under cognitive stress (*M* = 37.29, *SD* = 11.84) compared to without cognitive stress (*M* = 45.87, *SD* = 12.06), *t*(87) = 8.98, *p* < 0.001, Cohen’s *d* = 0.95. No other main effects or interactions were significant, *F*(1,87) ≤ 0.002, *p* > 0.97.

#### 3.2.2. Accuracy

A three-way mixed ANOVA on accuracy revealed a significant main effect of congruence, *F*(1,87) = 771.89, *p* < 0.001, *η_p_*^2^ = 0.90, with higher accuracy in congruent trials (*M* = 0.96, *SD* = 0.03) compared to incongruent trials (*M* = 0.77, *SD* = 0.07). No significant effects of cognitive stress, group, or interactions were found, *F*(1,87) ≤ 0.001, *p* > 0.98.

#### 3.2.3. Post-Error Slowing

Post-error slowing was analyzed by comparing the post-error and post-correct RTs. A two-way mixed ANOVA suggested that there were no significant main effects of cognitive stress, *F*(1,87) = 1.45, *p* = 0.23, *η_p_*^2^ = 0.02, or group, *F*(1,87) = 2.01, *p* = 0.16, *η_p_*^2^ = 0.02; no significant effect of interactions was found, *F*(1,87) = 2.15, *p* = 0.15, *η_p_*^2^ = 0.02.

### 3.3. Error-Related Brain Activity and Anxiety

Table 1 and Figure 2 provide descriptive and event-related potential statistics for ERN, CRN and ΔERN across conditions defined by cognitive stress and group.

#### 3.3.1. Three-Way ANOVA on ERN and CRN

The analysis revealed a significant main effect of component, *F*(1,87) = 131.65, *p* < 0.001, *η_p_*^2^ = 0.60, with ERN amplitudes (*M* = −2.74, *SD* = 3.14) being more negative than CRN amplitudes (*M* = 1.43, *SD* = 2.45). A main effect of group was also observed, *F*(1,87) = 4.83, *p* = 0.031, *η_p_*^2^ = 0.05, indicating overall larger amplitudes in the HTA group (*M* = −1.17, *SD* = 3.14) compared to the LTA group (*M* = −0.13, *SD* = 3.18). The interaction between component and group was marginally significant, *F*(1,87) = 3.44, *p* = 0.067, *η_p_*^2^ = 0.04. To further examine this effect, we conducted pairwise comparisons between groups for each component. ERN amplitudes showed a trend toward being more negative in the HTA group (*p* = 0.012; *M* = −3.59, *SD* = 3.14) than in the LTA group (*M* = −1.88, *SD* = 3.13), while CRN amplitudes did not differ between groups (*p* = 0.48).

#### 3.3.2. ΔERN

A two-way mixed ANOVA suggested that there was no significant main effect of cognitive stress on ΔERN, *F*(1,87) = 0.53, *p* = 0.47, *η_p_*^2^ = 0.006. The main effect of group was marginally significant, *F*(1,87) = 3.44, *p* = 0.067, *η_p_*^2^ = 0.04. No significant effect of their interaction on ΔERN was found, *F*(1,87) = 0.00, *p* = 0.99, *η_p_*^2^ < 0.001.

## 4. Discussion

To the best of our knowledge, prior studies on the impact of ERN have predominantly focused on its association with anxiety disorders. This study was the first to investigate the neurophysiological processes underlying the impact of test anxiety and cognitive stress on ERN. To this end, we created a test-like cognitive stress situation to examine the performance of individuals with test anxiety when performing a Flanker task in the absence of stress and under stressful conditions, and to assess changes in ERN amplitude. The results showed that HTA individuals tended to exhibit larger ERN amplitudes compared to LTA counterparts. However, ERN amplitudes did not significantly differ between pre- and post-stress conditions, suggesting that the stress manipulation did not have a significant impact on individuals’ stress responses.

[1] ([1]) found that individuals with larger ERN amplitudes are more likely to experience anxiety, particularly those exhibiting heightened sensitivity to internal threats, such as excessive worry or obsessive thoughts ([35]; [43]; [52]; [56]; [81]). Our observation aligned with these findings, as individuals with HTA demonstrated a trend toward larger ERN amplitudes during error monitoring. This pattern suggests that heightened error-related neural responses in HTA individuals may reflect increased sensitivity to errors. This enhanced error monitoring could be related to their excessive focus on internal threats, leading to a more intense reaction to errors ([53]).

In contrast, some research has shown that in younger populations, higher anxiety levels are associated with smaller ERN amplitudes, likely due to the immature functionality of the anterior cingulate cortex, which is crucial for error monitoring ([48]). Furthermore, young individuals with high anxiety often exhibited poorer cognitive control, including reduced flexibility in attention allocation and error monitoring, leading to weaker neural responses to error signals ([42]). This phenomenon is particularly evident in specific anxiety types, such as separation anxiety disorder, characterized by heightened emotional reactivity and inefficient cognitive control. As the nervous system matures, the relationship between anxiety and ERN undergoes significant changes. In mature individuals, particularly adults, the anterior cingulate cortex, a critical region for error monitoring, achieves full functionality, enhancing sensitivity to errors ([48]). This maturation allows individuals with higher anxiety levels to exhibit a tendency toward larger ERN amplitudes, reflecting enhanced error detection abilities and heightened sensitivity to internal threats. In our study, the participants were primarily adults with relatively mature cognitive and neural systems, including fully developed anterior cingulate cortex functionality. Consequently, their neural responses were broadly consistent with patterns typically reported in individuals with HTA, with a tendency toward larger ERN amplitudes during error monitoring.

In addition, prior research has suggested that ERN may serve as a neurobiological marker for anxiety disorders, reflecting heightened sensitivity to errors in individuals with anxiety ([47]). While our findings align with previous studies showing a tendency toward larger ERN amplitudes in HTA individuals, we did not observe a significant impact of stress manipulation on ERN amplitudes. Contrary to our expectations, ERN amplitudes did not differ significantly between pre- and post-stress conditions, indicating that the test-like stress situation did not have a substantial effect on individuals’ error monitoring responses. This suggested that ERN is not sensitive to the immediate stress induced by test-like situations in our sample, but instead reflects trait-level differences in test anxiety. In other words, the tendency toward larger ERN amplitudes observed in HTA individuals may be more indicative of their heightened sensitivity to errors as a trait of test anxiety, rather than a reaction to situational cognitive stress. This aligned with the notion that ERN could be a marker of individual differences in test anxiety rather than a response to external stressors. Additionally, previous research has linked ERN to maladaptive performance monitoring and hypervigilance to errors, which are characteristic of several anxiety disorders like generalized anxiety disorder, obsessive–compulsive disorder, and major depressive disorder ([31]; [35]). These observations support the idea of ERN as a potential transdiagnostic marker, but its relevance specifically to test anxiety should be further investigated.

This study successfully implemented a stress manipulation, evidenced by significant increases in both STAS and SSAI scores from pre- to post-stress and a significant main effect of the cognitive stress condition on RTs (*p* < 0.05). However, this increase in anxiety levels did not translate into significant changes in ERN amplitudes. One possible explanation for this finding is that cognitive stress may not significantly influence ERN. Previous studies have suggested that ERN is primarily driven by individual trait characteristics rather than state-dependent factors such as stress manipulation ([51]; [65]). For instance, [65] ([65]) reported that experimental stress was induced using the Trier Social Stress Test paradigm, followed by a go/no-go task to measure neural indices of error processing, particularly the ERN. The results showed that although the Trier Social Stress Test successfully induced physiological stress, as evidenced by increases in salivary cortisol, heart rate, and blood pressure, there were no significant differences in the ERN component measured during the go/no-go task between the stress and control groups. This indicated that experimental stress did not have a significant effect on error detection.

Similarly, [51] ([51]) investigated the effects of fear on error processing and attentional allocation using a spider fear induction paradigm. Spider-phobic individuals completed a Flanker task under both fear and control conditions, during which ERP components were measured. The results showed reduced amplitudes of the P300 and Pe components under the fear condition, indicating that fear diminished attentional allocation to task-relevant stimuli and reduced error salience. However, the ERN component was unaffected, suggesting that fear did not alter the early error processing stage. These findings align with the understanding of ERN as a stable neural response associated with error monitoring, which appears less sensitive to transient contextual changes. Another potential explanation lies in the nature of laboratory stress manipulations. Although our study emphasized score comparison, it may not have adequately simulated the multifaceted evaluative threats present in real-world test settings, such as the direct implications for future academic prospects. This limited realism may have contributed to the lack of significant stress effects on ERN amplitudes. The dissociation between neural and behavioral effects observed in our study highlights the differing sensitivities of neural and behavioral measures to stress manipulations. Previous studies have shown that ERN, as a neural response associated with error monitoring, may rely more heavily on realistic and sustained evaluative threat contexts. For instance, social evaluative threats have been shown to enhance ERN amplitudes, particularly among individuals with high punishment sensitivity, where ERN amplitudes correlate with physiological stress responses such as cortisol levels ([13]). Additionally, ERN changes reflect the degree to which errors are perceived as threatening, with greater perceived threat corresponding to larger ERN amplitudes ([82]). These findings collectively suggest that ERN changes may require more realistic stress conditions rather than simple laboratory instructions or short-term simulated stress manipulations. Future studies should aim to optimize the authenticity of stress manipulations to better elucidate the neural mechanisms underlying error monitoring.

Moreover, our study found no significant changes in CRN amplitudes in HTA and LTA individuals. This result suggested that while HTA individuals showed a tendency toward enhanced error monitoring reflected in ERN, this enhancement did not significantly affect self-monitoring during correct responses, as indicated by CRN. CRN primarily reflects self-monitoring activity during correct responses and is associated with overall performance monitoring demands. However, its changes are often smaller and less stable ([10]; [50]; [73]). For example, [63] ([63]) observed significantly enhanced CRN amplitudes in individuals with obsessive–compulsive disorder, a finding that may reflect their excessive focus on correct responses. In contrast, HTA individuals may focus more on the potential consequences of errors, resulting in less pronounced self-monitoring during correct responses. Additionally, individual differences may be another key factor contributing to the non-significant CRN results. Previous studies have shown that CRN amplitudes vary significantly across different traits or clinical populations. For instance, CRN did not show significant changes in children with generalized anxiety disorder ([36]), whereas it was significantly enhanced in individuals with obsessive–compulsive disorder ([63]). These differences suggested that CRN may more readily reflect specific disorders or cognitive styles, such as obsessive cognitive tendencies, which may not be as pronounced in the HTA population.

Above all, the core finding of this study is that test anxiety is associated with differences in ERN but not ΔERN. Importantly, the absence of a ΔERN effect is likely due to small CRN differences that reduced the ERN-CRN contrast, thereby diminishing the sensitivity of ΔERN to group differences. We believe the main reason for this pattern is that we observed a slight increase in CRN among HTA individuals compared to LTA individuals, which partially offset the effect of test anxiety on enhancing the ERN. Specifically, individuals with HTA tend to exhibit larger (more negative) ERN amplitudes than those with LTA, consistent with our hypotheses and previous findings ([15]; [34]; [46]). No significant group differences were observed for CRN and ∆ERN. However, the present study showed that the CRN in the HTA group was also more negative than that of the LTA group, although this difference did not reach statistical significance. This may be because, although previous studies found that anxiety did not significantly affect CRN ([69]; [75]), HTA individuals showed elevated CRN possibly due to enhanced motivation, which has been suggested to modulate CRN amplitude ([4]; [39]; [70]). However, this is only our speculation, and further research is needed to explore this issue. This trend toward more negative CRN may have partially offset the group difference in ERN, leading to a non-significant ∆ERN effect.

Despite the preliminary evidence provided by this study, one limitation warrants further investigation. While the stress manipulation employed in this study aimed to increase evaluative threat by introducing score comparisons, it was less impactful than real-world test settings, which often involve explicit consequences for academic outcomes or social evaluation. Our stress manipulation increased participants’ stress, but its intensity remained lower than that of real exams with graded outcomes and performance-ranking feedback. Such evaluative consequences in real testing situations may therefore elicit stronger neural responses. Future research should incorporate more realistic task designs, such as linking task performance directly to academic assessments, to better explore the role of stress in modulating ERN. Another limitation is the absence of a standardized trait anxiety assessment. Although trait and test anxiety are correlated, they are conceptually distinct ([41]; [86]). In student populations, test anxiety is the most prominent form of anxiety, and thus it was our primary focus. Future studies are recommended to incorporate a standard trait measure (e.g., STAI-T) to clarify the relative contribution of each construct. Additionally, the cross-sectional design limits causal inferences, preventing us from determining whether enhanced error monitoring is a precursor to or a consequence of test anxiety. Moreover, future studies should incorporate physiological markers such as cortisol, alpha-amylase, HRV, and skin conductance to more directly validate the activation of HPA and SAM stress systems. Such measures, together with indices like AUCi and AUCg and with time-locked sampling, would help clarify whether the stress manipulation was sufficiently potent to explain the dissociation between behavioral and neural responses.

This study was the first to examine the impact of test anxiety and cognitive stress on ERN. Our findings revealed that HTA individuals exhibit a trend toward enhanced ERN amplitudes, reflecting their heightened sensitivity to errors and increased error monitoring capability. However, contrary to expectations, stress conditions did not significantly alter ERN amplitudes, suggesting that laboratory-based evaluative threat manipulations may require higher authenticity to elicit significant neural effects. These findings highlight the complexity of stress contexts in modulating error monitoring among individuals with test anxiety. The study expanded the understanding of the relationship between test anxiety and error monitoring mechanisms, further supporting the potential of ERN as a biomarker for anxiety. From a practical perspective, the findings provided valuable insights into the early identification and intervention for test anxiety. For example, ERN-based neurophysiological assessments could help identify high-risk individuals and inform targeted interventions, such as cognitive behavioral therapy or mindfulness training, to mitigate the negative impact of test anxiety on academic performance.

## 5. Conclusions

While the stress manipulation effectively increased subjective anxiety, it did not produce significant changes in ERN, CRN, or ΔERN amplitudes. These findings suggest that ERN amplitude is more strongly influenced by trait test anxiety than by transient cognitive stress.

## Figures and Tables

**Figure 2 behavsci-16-00025-f002:**
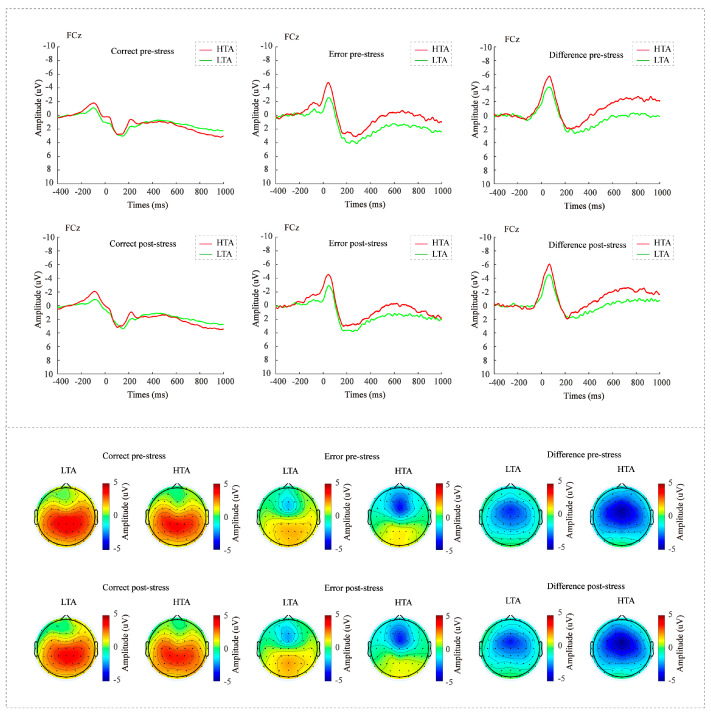
The upper panel shows the response-locked ERP waveforms at FCz for CRN, ERN, and ΔERN in the pre-stress and post-stress conditions. The lower panel shows the topographies (0–100 ms) in the pre-stress and post-stress conditions. *Notes:* HTA = high test anxiety; LTA = low test anxiety.

**Table 1 behavsci-16-00025-t001:** Demographic, Scale, behavioral, and brain potential data in high test anxious (HTA) and low test anxious (LTA).

	LTA (N = 44)		HTA (N = 45)	
	Mean	SD	Mean	SD
Demographic data				
Age (year)	19.82	1.42	19.53	1.27
Sex (female)	N = 23		N = 25	
Scale checklist				
TAS	8.57	2.43	27.49	3.51
Pre STAS	1.93	0.82	2.67	0.74
Pre SSAI	9.95	3.07	13.11	3.00
Post STAS	2.11	0.87	3.00	0.77
Post SSAI	11.73	3.39	14.91	3.35
Behavioral data				
Pre				
Acc	0.86	0.05	0.86	0.07
Acc on C trials	0.96	0.04	0.95	0.05
Acc on IC trials	0.77	0.08	0.76	0.10
Acc after correct trials	0.87	0.05	0.87	0.06
Acc after incorrect trials	0.86	0.11	0.84	0.14
Error RT (ms)	389.66	34.94	389.70	39.29
Correct RT (ms)	441.00	41.80	443.60	37.37
RT on C trials (ms)	411.26	39.80	413.41	35.15
RT on IC trials (ms)	457.36	43.80	459.04	40.19
Post-error slowing (ms)	10.52	15.42	4.64	13.81
Post				
Acc	0.86	0.04	0.86	0.04
Acc on C trials	0.96	0.03	0.96	0.02
Acc on IC trials	0.77	0.07	0.76	0.07
Acc after correct trials	0.86	0.04	0.86	0.04
Acc after incorrect trials	0.88	0.07	0.86	0.09
Error RT (ms)	371.56	33.54	373.88	31.26
Correct RT (ms)	415.53	46.19	417.50	32.34
RT on C trials (ms)	391.83	44.96	392.46	28.81
RT on IC trials (ms)	428.78	48.62	430.08	35.99
Post-error slowing (ms)	10.14	16.57	8.47	10.64
Event-related brain potential data				
Pre				
ERN, FCz (μV)	−1.77	2.41	−3.65	4.18
CRN, FCz (μV)	1.66	2.14	1.12	2.63
ΔERN, FCz (μV)	−3.42	2.90	−4.77	4.13
Post				
ERN, FCz (μV)	−1.99	3.00	−3.53	3.49
CRN, FCz (μV)	1.57	3.06	1.36	2.31
ΔERN, FCz (μV)	−3.56	3.25	−4.90	3.68

*Notes:* TAS = Test anxiety scale; STAS = Subjective Test Anxiety Scale; SSAI = Short State Anxiety Inventory; IC = incongruent condition; C = congruent condition; RT = reaction time; Acc = accuracy; ERN = Error-related negativity; CRN = Correct response negativity; ΔERN = ERN − CRN.

## Data Availability

Dataset available on request from the authors.

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
