# Peer review of "The Impact of Test Anxiety and Cognitive Stress on Error-Related Brain Activity"

_behavsci, 2025, doi:10.3390/bs16010025_

Round 1
Reviewer 1 Report
Comments and Suggestions for Authors
This is a well-designed and clearly presented study examining the relationship between test anxiety, cognitive stress, and error-related brain activity. The methodology is appropriate and the conclusions align with the results. The manuscript contributes to clarifying the distinction between trait-like test anxiety effects on ERN and state-induced stress effects.
Although subjective anxiety increased, the evaluative stress condition may not fully capture real exam pressure. Consider noting in the Limitations that stronger real-life consequences (e.g., graded outcomes, performance ranking feedback) may produce more robust neural modulation.
Including a standard measure of trait anxiety in future work (e.g., STAI-T) would help separate test-specific anxiety from general anxiety sensitivity.
Since ΔERN did not differ by group, clarify that this may be related to the small non-significant CRN differences, which can reduce the ΔERN contrast.
Consider incorporating physiological validation of the stress induction by measuring both HPA (salivary cortisol at baseline, pre-task, +20 min, +35–40 min) and SAM (salivary alpha-amylase at baseline, immediate post-stress, +5–10 min; plus HR/HRV and skin conductance during task). Report AUCi/AUCg for cortisol and time-lock SAM metrics to the RSPM/Flanker periods. These additions would objectively confirm engagement of stress systems and help interpret the null ERN change versus clear behavioral speeding under stress.
Author Response
Authors’ replies
Manuscript ID:behavsci-3924901
Title: The impact of test anxiety and cognitive stress on error-related brain activity
Comments 1: This is a well-designed and clearly presented study examining the relationship between test anxiety, cognitive stress, and error-related brain activity. The methodology is appropriate and the conclusions align with the results. The manuscript contributes to clarifying the distinction between trait-like test anxiety effects on ERN and state-induced stress effects.
Response 1: Thank you very much for your positive and encouraging comments. We sincerely appreciate your recognition of the study design, methodology, and clarity of presentation. Your feedback has been very motivating for us. All revisions addressing this comment have been marked in red in the main text.
Comments 2: Although subjective anxiety increased, the evaluative stress condition may not fully capture real exam pressure. Consider noting in the Limitations that stronger real-life consequences (e.g., graded outcomes, performance ranking feedback) may produce more robust neural modulation.
Response 2: Thank you for this valuable comment. We fully agree that the evaluative stress induced in our paradigm may not reflect the full intensity of real-world exam pressure. Accordingly, we have revised the Limitations section to explicitly acknowledge that stronger real-life consequences, such as graded outcomes or performance ranking feedback, may produce more robust neural modulation than the current laboratory-based manipulation. This revision has been made on page 13, paragraph 3, lines 540-543.
Comments 3: Including a standard measure of trait anxiety in future work (e.g., STAI-T) would help separate test-specific anxiety from general anxiety sensitivity.
Response 3: We appreciate the reviewer’s suggestion regarding the inclusion of a standard trait anxiety measure. In response, we have added a statement in the Limitations section recommending that future studies incorporate validated trait anxiety instruments (e.g., STAI-T) to better distinguish test-specific anxiety from general anxiety sensitivity. This revision has been added on page 13, paragraph 3, lines 548–550.
Comments 4: Since ΔERN did not differ by group, clarify that this may be related to the small non-significant CRN differences, which can reduce the ΔERN contrast.
Response 4: Thank you for this helpful suggestion. We agree that the absence of a group difference in ΔERN may be influenced by the small and non-significant differences in CRN, which would reduce the subtraction-based ERN-CRN contrast and weaken the ΔERN effect. We have now made this explanation more explicit in the Discussion to ensure clearer interpretation of the result. The revised text is provided on page 13, paragraph 2, lines 518-521.
Comments 5: Consider incorporating physiological validation of the stress induction by measuring both HPA (salivary cortisol at baseline, pre-task, +20 min, +35–40 min) and SAM (salivary alpha-amylase at baseline, immediate post-stress, +5–10 min; plus HR/HRV and skin conductance during task). Report AUCi/AUCg for cortisol and time-lock SAM metrics to the RSPM/Flanker periods. These additions would objectively confirm engagement of stress systems and help interpret the null ERN change versus clear behavioral speeding under stress.
Response 5: We thank the reviewer for this insightful suggestion. We fully agree that simultaneously assessing HPA-axis (salivary cortisol: baseline, pre-task, +20 min, +35–40 min) and SAM-system (salivary α-amylase: baseline, immediate post-stress, +5–10 min; HR/HRV and electro-dermal activity throughout the task) would provide objective verification of stress-system engagement and help clarify why stress produced robust behavioral speeding but no ERN modulation. Although these measures were not available in the present dataset, we now explicitly acknowledge this limitation and outline the above protocol as a direct extension for future studies (page 13-14, lines 552-557). Implementing such multimodal validation would allow us to compute AUCi/AUCg for cortisol and time-lock SAM reactivity to the RSPM/Flanker periods, thereby testing whether null ERN effects reflect insufficient HPA activation, rapid SAM habituation, or dissociable neuro-cognitive mechanisms.
Reviewer 2 Report
Comments and Suggestions for Authors
Dear authors.
Thank you for submitting your article to the journal. The work under consideration is of considerable interest, and the subject matter is of current relevance. Nevertheless, I would like to offer some minor suggestions for its improvement.
- Introduction
/ The flow of the introduction could be improved. At times, the transition between ideas feels abrupt. I suggest reordering the paragraphs to create a more logical progression
/ The obsessive-compulsive disorder is mentioned in text, but the rationale for its inclusion is not fully explained.
/ The gap in literature should be more emphasized. If you work is pioneer in this field, you can highlight it.
/ The hypothesis should be stated in more technical terms.
- Method
/ Include a sample item for the in TAS measure.
/ Please, include the age range of the participants and state the period of data collection.
- Results
/ In line 316 the statistic "F(1,86) = 0.00, p = 1.00" suggests a potential reporting error or rounding issue. Please verify.
/ In line 342-3, the M and SD statistics should be italicized.
/ In line 444, the p statistic should be italicized.
/ In Section 3.3.1, the interaction between Component and Group was not significant (p = .067). Following up a non-significant interaction with pairwise comparisons is generally discouraged. The authors should tone down the conclusion that HTA significantly differs from LTA based on this ANOVA, or provide a stronger justification for analyzing simple effects (e.g., a priori hypotheses)
- Discussion
/ The opening paragraph states that HTA individuals exhibited "significantly larger ERN amplitudes." Given that the interaction effect in the Results was marginally significant (p = .067), this language is too strong. Please rephrase to reflect the statistical reality (e.g., "a trend toward larger amplitudes") to avoid misleading the reader.
/ In limitations section, you can note the cross-sectional comparison of the anxiety groups.
- References
/ The references must align with the journal´s guidelines. Please, review it.
/ In line 596, “anxiety” appears as an author. Check this.
/ In line 622, “emotion” appears as an author. Check this.
I hope these comments are helpful.
Best regards.
Author Response
Authors’ replies
Manuscript ID:behavsci-3924901
Title: The impact of test anxiety and cognitive stress on error-related brain activity
Dear authors.
Comments 1: Thank you for submitting your article to the journal. The work under consideration is of considerable interest, and the subject matter is of current relevance. Nevertheless, I would like to offer some minor suggestions for its improvement.
Response 1: Thank you very much for your positive evaluation and constructive comments. I appreciate your recognition of the relevance and value of this work. I have carefully considered the suggestions and will make the corresponding revisions to further improve the clarity and quality of the manuscript. All revisions addressing this comment have been marked in red in the main text.
- Introduction
Comments 2: The flow of the introduction could be improved. At times, the transition between ideas feels abrupt. I suggest reordering the paragraphs to create a more logical progression
Thank you for this valuable suggestion. We agree that the introduction would benefit from reordering to improve coherence and strengthen the logical flow. We have substantially reorganized the text into five sequential steps: (1) introduction of test anxiety, (2) explanation of the ERN as a relevant neural marker, (3) conflicting findings in the literature, (4) the key debate regarding trait-like versus situational influences, and (5) the resulting gap that motivates the present study. Accordingly, we relocated the paragraph that introduces the ERN to page 2, paragraph 1, lines 41-55, so that readers encounter the neural mechanism before the empirical findings. In addition, we inserted brief transitional sentences to bridge sections and eliminate abrupt shifts. We believe these revisions significantly enhance the clarity and readability of the introduction.
Comments 3: The obsessive-compulsive disorder is mentioned in text, but the rationale for its inclusion is not fully explained.
Response 3: Thank you for this helpful comment. We agree that the inclusion of obsessive-compulsive disorder needed further contextualization. In the revised Introduction, we have clarified that obsessive-compulsive disorder serves as a representative clinical example within the anxiety spectrum. Specifically, we referenced obsessive-compulsive disorder because it shares key phenomenological features with test anxiety, namely, excessive worry and hyperactive performance monitoring. Thus, the finding that obsessive-compulsive disorder patients exhibit elevated ERN even without stress provides critical support for the “trait-based” perspective. The added sentences are “To exemplify this, obsessive-compulsive disorder has often been cited as a representative clinical example within the anxiety spectrum. We reference obsessive-compulsive disorder here because, much like test anxiety, it is characterized by excessive worry and hyperactive performance monitoring”. The revised text is provided on page 3, paragraph 2, lines 105-109.
Comments 4: The gap in literature should be more emphasized. If you work is pioneer in this field, you can highlight it.
Response 4: Thank you for pointing out the need to emphasize the literature gap more clearly. We agree with this suggestion and have strengthened the introduction by explicitly stating what is currently unknown in the field. Specifically, on page 3, paragraph 3, lines 127-130, we added sentences that state that “Despite extensive ERN research in clinical and trait anxiety, to the best of our knowledge, no study has empirically tested this specific interaction within the domain of test anxiety. Consequently, it remains unexplored whether these neural mechanisms are driven by the trait itself or by situational stress.”
Comments 5: The hypothesis should be stated in more technical terms.
Response 5: Thank you for this helpful suggestion. We agree that the hypothesis would benefit from more technical precision, and we have revised its wording accordingly. Specifically, on page 4, paragraph 1, lines 143-145, we now state that “we hypothesized that HTA individuals would show enhanced ERN amplitudes relative to LTA individuals, and that this ERN increase would be significantly larger under the cognitive-stress condition.” This revision improves the technical accuracy of the hypothesis and presents it in a form that more clearly reflects neurophysiological prediction.
- Method
Comments 6: Include a sample item for the in TAS measure.
Response 6: Thank you for this helpful suggestion. We agree that including a sample item of the TAS enhances the clarity of the Method section. Accordingly, we have inserted the first item of the scale to illustrate the specific content rated by participants. The revised sentence reads: “For example, Item 1 reads: “Before an important test, I often think others are much smarter than I am,” to which participants respond either agree or disagree.” This revision can be found on page 4, paragraph 3, lines 182-185.
Comments 7: Please, include the age range of the participants and state the period of data collection.
Response 7: Thank you for your suggestion. We have now added both the age range of participants and the period of data collection to the Participants section. This information has been inserted on page 4, paragraph 2, lines 164-165, where we state that the participants were aged 18-23 years, and that data were collected from March 1 to May 11, 2024. We believe this addition improves methodological transparency and fully addresses the reviewer’s request.
- Results
Comments 8: In line 316 the statistic "F(1,86) = 0.00, p = 1.00" suggests a potential reporting error or rounding issue. Please verify.
Response 8: Thank you for your careful observation. We have re-verified the SPSS ANOVA output and confirm that the reported statistic is correct. The result F(1, 86) = 0.00 with p = 1.00 reflects that this effect accounted for no explainable variance in the model, yielding a mean square of zero for that term. This interpretation is further supported by the extremely small effect size (ηp² < .001). We therefore retained the exact statistical values as reported by the software.
Comments 9: In line 342-3, the M and SD statistics should be italicized.
Response 9: Thank you for noticing this formatting detail. The M and SD statistics on line 358 have now been italicized in accordance with APA style requirements.
Comments 10: In line 444, the p statistic should be italicized.
Response 10: We appreciate this correction. The p value on line 459 has been updated to italic formatting to ensure statistical reporting consistency throughout the manuscript.
Comments 11: In Section 3.3.1, the interaction between Component and Group was not significant (p = .067). Following up a non-significant interaction with pairwise comparisons is generally discouraged. The authors should tone down the conclusion that HTA significantly differs from LTA based on this ANOVA, or provide a stronger justification for analyzing simple effects (e.g., a priori hypotheses)
Response 11: Thank you for the comment. As the interaction was marginal, we have toned down the interpretation and removed the word significantly from the conclusion. We retained the pairwise comparison because group differences in ERN were part of our prespecified hypothesis, but we now report this result as a trend rather than a confirmed effect. This revision has been made on page 10, paragraph 3, line 392.
- Discussion
Comments 12: The opening paragraph states that HTA individuals exhibited "significantly larger ERN amplitudes." Given that the interaction effect in the Results was marginally significant (p = .067), this language is too strong. Please rephrase to reflect the statistical reality (e.g., "a trend toward larger amplitudes") to avoid misleading the reader.
Response 12: Thank you for this crucial correction. We agree that, given the marginal interaction (p = 0.067), the use of “significantly” was imprecise and potentially misleading. We have revised the opening paragraph and carefully reviewed the entire manuscript to ensure the language accurately reflects the statistical reality. We have replaced strong assertions with phrases such as “tended to exhibit” or “showed a trend toward” throughout the Results, Discussion, and Conclusion sections. For instance, the phrase “significantly larger ERN amplitudes” has been updated to “tended to exhibit larger ERN amplitudes.” These comprehensive revisions can be found on page 11, paragraph 1, line 407, as well as in the Discussion on page 11, paragraph 2, lines 415-417, page 11, paragraph 3, lines 431-432, page 11, paragraph 3, lines 435-437, page 11, paragraph 4, line 447, page 12, paragraph 4, lines 500-501 and page 13, paragraph 2, line 524.
Comments 13: In limitations section, you can note the cross-sectional comparison of the anxiety groups.
Response 13: We appreciate the reviewer’s suggestion. We have now added a statement acknowledging that the cross-sectional comparison between HTA and LTA groups limits causal inferences regarding the directionality of the relationship. This clarification has been added on page 13, paragraph 3, lines 550-552.
- References
Comments 14: The references must align with the journal´s guidelines. Please, review it.
Response 14: Thank you for noting the need to ensure full alignment with the journal’s reference format. We have carefully reviewed and revised the reference list to meet the journal’s formatting specifications. The updated reference list now conforms to the required guidelines.
Comments 15: In line 596, “anxiety” appears as an author. Check this.
Response 15: Thank you for pointing out this formatting error. The reference in line 619 incorrectly listed “anxiety” as the author due to a citation import issue. We have now corrected the entry to reflect the proper author name according to journal guidelines.
Comments 16: In line 622, “emotion” appears as an author. Check this.
Response 16: Thank you for your careful observation. The reference in line 646 mistakenly listed “emotion” as the author due to a citation formatting error. We have corrected this reference to display the appropriate author name in accordance with the journal’s style requirements.
Comments 17: I hope these comments are helpful.
Best regards.
Response 17: Thank you again for your thoughtful and constructive comments. They have been very helpful in improving the quality of our manuscript.
Round 2
Reviewer 2 Report
Comments and Suggestions for Authors
Dear authors.
I hope this message finds you well.
Thank you for considering my previous comments. I think that your article has significantly improved. I would like to congratulate you on the work done.
Your article is suitable for publication. However, there are some minor mistakes in References that should be corrected to improve the overall quality of your work.
/ References:
- Some journal names are missing and there are mistakes in the authors. For example, in the reference “Xu, C., & Wei, H. (2024)”, the journal is missing and “J. B. p.” is not correct.
- In the first reference (Amir et al., 2024), you can add all authors.
- In Fei and Jian-xing (2008) reference, the page number is not correct.
- In Rodeback et al. (2020) reference, the numbers “14, 16, 189” are not clear. What is volume and what page number?
- Please check carefully the References section.
I wish you a Merry Christmas and a Happy New Year.
Best regards.
Author Response
Authors’ replies
Manuscript ID:behavsci-3924901
Title: The impact of test anxiety and cognitive stress on error-related brain activity
Dear authors.
I hope this message finds you well.
Comments 1: Thank you for considering my previous comments. I think that your article has significantly improved. I would like to congratulate you on the work done.
Your article is suitable for publication. However, there are some minor mistakes in References that should be corrected to improve the overall quality of your work.
Response 1: Thank you very much for your kind message and for your positive evaluation of our revised manuscript. We sincerely appreciate your recognition that the article has significantly improved and your conclusion that it is suitable for publication. Your encouragement is greatly appreciated.
/ References:
Comments 2: Some journal names are missing and there are mistakes in the authors. For example, in the reference “Xu, C., & Wei, H. (2024)”, the journal is missing and “J. B. p.” is not correct.
Response 2: Thank you for carefully checking the References section. We have added the missing journal information and corrected the author entry for Xu, C., and Wei, H. (2024). The incorrect abbreviation “J. B. p.” has also been removed. These corrections can be found on page 19, lines 788-789.
Comments 3: In the first reference (Amir et al., 2024), you can add all authors.
Response 3: Thank you for this suggestion. We have now provided the complete author list for Amir et al. (2024) to improve accuracy and completeness. This revision has been made on page 15, lines 601-603.
Comments 4: In Fei and Jian-xing (2008) reference, the page number is not correct.
Response 4: Thank you for pointing this out. We have corrected the page number information for Fei and Jian-xing (2008). The corrected reference appears on page 16, lines 663-664.
Comments 5: In Rodeback et al. (2020) reference, the numbers “14, 16, 189” are not clear. What is volume and what page number?
Response 5: We appreciate the reviewer’s careful attention to this reference. We have clarified the volume and article-number information for Rodeback et al. (2020) according to the journal’s official publication details. This correction can be found on page 18, lines 748-750.
Comments 6: Please check carefully the References section.
Response 6: In response to this comment, we carefully reviewed the entire References section and corrected any remaining inconsistencies to ensure full compliance with the journal’s guidelines. We would also like to note that some journals do not use issue numbers and therefore report only the volume with page ranges, while others do not assign page numbers and instead use article numbers. In such cases, the references have been formatted according to the official publication information provided by the respective journals.
Comments 7: I wish you a Merry Christmas and a Happy New Year.
Best regards.
Response 7: Thank you again for your valuable feedback and for helping us improve the quality of our manuscript. We wish you a Merry Christmas and a Happy New Year as well.